# Automated Crack Detection in Monolithic Zirconia Crowns Using Acoustic Emission and Deep Learning Techniques

**DOI:** 10.3390/s24175682

**Published:** 2024-08-31

**Authors:** Kuson Tuntiwong, Supan Tungjitkusolmun, Pattarapong Phasukkit

**Affiliations:** 1Department of Biomedical Engineering, School of Engineering, King Mongkut’s Institute of Technology Ladkrabang, Bangkok 10520, Thailand; kuson.tu@kmitl.ac.th (K.T.); supan.tu@kmitl.ac.th (S.T.); 2King Mongkut Chaokhun Thahan Hospital, King Mongkut’s Institute of Technology Ladkrabang, Bangkok 10520, Thailand

**Keywords:** monolithic zirconia dental crown, crack, convolutional neural network (CNN), deep learning, inception-resnet-v2, acoustic emission (AE), pencil lead breaking (PLB), continuous wavelet transform (CWT)

## Abstract

Monolithic zirconia (MZ) crowns are widely utilized in dental restorations, particularly for substantial tooth structure loss. Inspection, tactile, and radiographic examinations can be time-consuming and error-prone, which may delay diagnosis. Consequently, an objective, automatic, and reliable process is required for identifying dental crown defects. This study aimed to explore the potential of transforming acoustic emission (AE) signals to continuous wavelet transform (CWT), combined with Conventional Neural Network (CNN) to assist in crack detection. A new CNN image segmentation model, based on multi-class semantic segmentation using Inception-ResNet-v2, was developed. Real-time detection of AE signals under loads, which induce cracking, provided significant insights into crack formation in MZ crowns. Pencil lead breaking (PLB) was used to simulate crack propagation. The CWT and CNN models were used to automate the crack classification process. The Inception-ResNet-v2 architecture with transfer learning categorized the cracks in MZ crowns into five groups: labial, palatal, incisal, left, and right. After 2000 epochs, with a learning rate of 0.0001, the model achieved an accuracy of 99.4667%, demonstrating that deep learning significantly improved the localization of cracks in MZ crowns. This development can potentially aid dentists in clinical decision-making by facilitating the early detection and prevention of crack failures.

## 1. Introduction

Monolithic zirconia (MZ) crowns have become widely popular as the material of choice for dental crowns, especially for restoring severely damaged teeth [1,2,3]. High-quality aesthetics, biocompatibility, and mechanical strength are recognized attributes of MZ crowns. As a result, MZ has been gradually replacing conventional ceramic systems as the favored restorative material. A critical factor in the processing of MZ is the development of computer-aided design and computer-aided manufacturing (CAD/CAM). Dental crowns for root canal-treated teeth and implant prostheses, as well as dental bridges, can all be fabricated using CAD/CAM technology, which has gained widespread use. As such, to reduce the chance of failure, it is essential to identify and localize cracks in these crowns as soon as possible. Furthermore, mechanical loading and structural problems can lead to crown failures. MZ crowns are fabricated using a multi-step, intricate process that can result in a variety of surface flaws [4,5], particularly when subjected to extreme stresses such as those brought on by bruxism [6,7,8]. Currently, several nondestructive methods are available for detecting flaws, including direct visualization [9], fiber-optic transillumination, combined tactile examination [9], radiographic examination, computer tomography [10,11], and acoustic emission (AE) testing [12]. In clinical evaluations, dental crown removal is not feasible, whereas destructive testing methods, such as Vickers indentation and fractography, are limited to laboratory settings [13,14]. The subtle nature of cracks in dental crowns render such defects challenging to detect, leading to difficulties in diagnosis by clinicians. Misdiagnosis can result from visual fatigue or judgment errors. Flaws may be missed if the X-ray beam does not align with the defect, as seen with cone-beam computed tomography (CBCT), which is prone to streaking artifacts from radiopaque materials [11]. With voxel sizes of 75–400 μm, CBCT often fails to detect finer microcracks [15]. Noise and interference are nearly unavoidable, and limited resolution may obscure microcracks, especially in early detection. Micro-computed tomography (micro-CT) often suffers from artifacts that degrade image quality, and its high resolution requires radiation doses in SkyScan 1272 workflow of up to 141.4 kVp, far exceeding the clinical CBCT 60–90 kVp [16]. To combat such disadvantages, AE analysis emerges as a potential alternative method. AE testing is a widely used passive nondestructive testing (NDT) method that evaluates stress waves generated by cracking within a material [17]. It detects the spontaneous release of strain energy from microscopic or macroscopic changes, converting it into an electrical signal for further analysis [18]. These emissions can be released from localized stress concentrations, sudden material changes, crack propagation, or other dynamic processes. As a nondestructive technique, AE testing measures and examines the elastic energy released during crack formation. AE testing is frequently employed in structural health monitoring (SHM) to assess the integrity of storage tanks [19,20], railways [21], aircraft parts [22,23,24,25], welding processes [26], medical devices such orthopedic joint prostheses [27,28], and dental materials [29,30,31,32,33].

SHM therefore offers a more dependable and affordable method for the maintenance of dental restorations, as it enables early detection and prevention of cracks that may impact the structural integrity of dental crowns. Accurate fracture localization and analysis are crucial for improving the routine practical inspections of dental restorations. Noninvasive techniques in AE technologies, known as modalities, are preferred due to their nondestructive nature. AE is particularly effective at isolating individual components [19,34]. The ability of AE testing to monitor real-time debonding kinetics at the tooth-composite/prosthesis interface offers valuable insights into the effects of interfacial debonding [35,36,37,38,39]. AE testing can help identify damage progression during fatigue tests by analyzing location, energy, frequency, and waveform direction [40]. Therefore, we propose that AE testing can prove to be a superior technology for dental material analysis. High-resolution spatial signals with denoised information can be used to develop automated diagnostic tools for dental practitioners to identify and analyze cracks with higher efficiency and accuracy. Early diagnosis and classification of faults in restorations are essential to improve treatment outcomes and increase the survival rate of dental restorations. At the same time, artificial intelligence-based methods can provide quick decision-making without human intervention, and deep learning (DL) holds great potential for the evaluation of zirconia combined with nondestructive testing–acoustic emission (NDT-AE) [18]. Crack recognition in dentistry can be challenging and tends to rely heavily on automated diagnostic systems. This study focuses on identifying cracks on crown surfaces during clinical examination, with the aim of developing sophisticated techniques for the classification of cracks. A study by Lee, D.W. et al. showed that the automated CNN architecture using radiographic image analysis techniques had both limitations and future potential. All three DL algorithms achieved an AUC of over 0.90 in detecting fractured dental implants, with the automated Deep Convolutional Neural Network using periapical images outperforming both the modified VGGNet-19 and GoogLeNet Inception-v3 [41]. The advancement of technology has led to the development of acoustic emission (AE) detection for restorations, which can identify the position of cracks. Automated or computerized methods are expected to reduce manual diagnostic errors and provide precise data for accurate diagnosis and treatment planning. Current crack detection methods transform AE signals to continuous wavelet transform (CWT) for more accurate result interpretation. AE-NDT employing CWT images allows for multi-class classification to precisely identify crack locations, particularly those close to adjacent teeth in the oral cavity. This study investigated crack detection at five vulnerable areas of the crown: incisal (cutting edge), left, right, labial (anterior), and palatal (posterior) surfaces.

This study utilized the Enhanced Inception-ResNet-v2 model, combining the advantages of the Inception and ResNet models. This model employs multi-scale depth-wise separable convolution for efficient and fast feature extraction, and includes a feature extractor to preserve important formation, accelerating convergence and filtering out less relevant data. The model eliminates redundant parameters to maintain controllable complexity and reduce computational load. Compared to existing algorithms, Inception-ResNet-v2 demonstrates superior accuracy and stability [42]. For image classification tasks, the Inception-ResNet-v2 model that is suggested has been shown to be quite effective. The model handles multiple scales of information from various input images, aiding in diagnosing diseases. By leveraging the combined strengths of Inception and ResNet architectures, the model enhances training scope, robustness, and depth, achieving superior efficiency and stability. The Inception-ResNet-v2 model demonstrates higher classification accuracy than other existing methods, making it an effective tool in machine learning for image classification problems [43]. This research aimed to enhance crack detection and localization in MZ crowns utilizing CWT to generate 2D images from AE signals. These images were analyzed using CNNs to analyze the AE-CWT patterns, specifically the Inception-ResNet-v2 architecture. CWT preprocessing was applied to raw AE signals. Images were processed with AE technology produced by pencil lead breaking (PLB) to classify crack sites in MZ crowns into one of the five specified categories. CNNs automatically extracted features through various convolutional layers, addressing the challenges of crack classification without manual split regions and feature extraction. Consequently, the extracted features were expected to yield robust results for subsequent classification. Utilizing the Inception-ResNet-v2 architecture, the AE-CWT patterns were analyzed to localize and classify cracks into one of five specified categories. This research aimed to contribute to the dental field by providing a classification system for cracks in zirconia crowns, thereby facilitating their identification and management in clinical practice.

This research is organized as follows: the introduction and review of several earlier studies that used the AE and DL models for the classification of dental material detection are included in Section 1. The data and methodology utilized in this work are described in Section 2, with a particular emphasis on the deep learning architecture. The Inception-ResNet-v2 analysis results are shown in Section 3, accompanied by an assessment of the effectiveness of the crack classification system. Section 4 presents the discussion. The conclusion of this study and recommendations for future research are provided in Section 5.

## 2. The Proposed AE-Based Damage Localization Method

This experiment aimed to classify and localize crack defects in MZ dental crowns using the AE-CWT transform. The proposed methodology investigated the application of CWT methods to extract key AE components, combined with the crack feature classification capabilities of a DL algorithm. AE signals indicative of cracks were initially identified, and their scalograms were generated using the CWT method. The CWT of a given signal *x*(*t*) is defined by Equation (1) [44,45].
(1)$${CWT}_{x}^{\psi }(a,b)=\Psi_{x}^{\psi }(a,b)=\frac{1}{\sqrt{|a|}}\int_{-\infty }^{\infty }x(t)\psi^{*}(\frac{t-b}{a})dt$$
where *x*(*t*) is the input signal at time (*t*), and *ψ*(*t*) is the analyzing wavelet, a small, localized waveform. The scale parameter *a* adjusts the width of the wavelet function, while the translation parameter *b* shifts the wavelet function along the time axis. Both *a*, *b*
ϵ R and the term *** denote the complex conjugate of the wavelet function.

### 2.1. Architecture of the Network

The Inception-ResNet-v2 architecture [42] was employed for MZ crack classification. This CNN, a variation of Inceptionv3, is 164 layers deep and combines the Inception architecture with residual connections. The architecture of the network, depicted in Figure 1, comprised three parts: the Stem (Figure 2a), where the original input was preprocessed using deep convolutional layers in this portion before entering the Inception-ResNet blocks. It included nine convolutional layers and two max-pooling layers. The second section consisted of Inception-ResNet blocks, detailed in Figure 2, and comprised the following blocks: (1) Inception-ResNet-A (Figure 2b), which featured an inception module with two 3 × 3 kernels; (2) Inception-ResNet-B (Figure 2c), which incorporated an asymmetric filter combination using one 1 × 7 filter and one 7 × 1 filter; (3) Inception-ResNet-C (Figure 2d), which featured a filter combination of one 1 × 3 filter and one 3 × 1 filter, which were small and asymmetric, with the 1 × 1 convolutions preceding the larger filters. Reduction modules A and B (Figure 2e,f) enhanced dimensionality to compensate for the reduction that the Inception blocks initially created. By using asymmetric convolution splitting, the architecture of the network increased the variety of filter patterns available. The pooling and softmax layers was included in the prediction layer, which was the final section. The feature maps of the Stem, Inception-ResNet-A, B, and C modules were combined by the multi-scale context information fusion module to facilitate the detection of small cracks. The network design effectively incorporated residual connections to address deep architecture challenges, and the inclusion of diverse filter patterns enhanced its representation capabilities.

### 2.2. Methodology Proposal

The proposed methodology evaluated the effectiveness of CWT methods for extracting key AE components and the capability of a DL algorithm for classifying crack features. The identified AE signals indicative of cracks were transformed into scalograms using the CWT method. A 2D-CNN model for automated damage detection and localization subsequently processed these scalograms. Specification: the scalograms served as input for the 2D-CNN, which employed an Inception-ResNet-v2 architecture to classify and localize cracks. The overview methodology is visually outlined in the flowchart depicted in Figure 3. In this study, the architecture referred to as Inception-ResNet-v2 combined elements from the Inception block and the ResNet structure [42]. This combined architecture was utilized to classify the CWT images derived from AE data.

### 2.3. Setup of the AE Data Acquisition System

#### Experimental Procedure

The experiment was conducted as part of clinical material research aimed at detecting and analyzing the location of cracks in MZ dental crowns. A comprehensive crack detection test was arranged to perform AE measurements. The experimental setup was designed to test surface cracks in MZ crowns composed of five sides and clinically designed for dental applications. The elemental composition of the MZ crown included the main component (ZrO_2_), with proportions of 0.05–1% aluminum. In addition, 3–12% yttria was used as an alternative stabilizer [46].

### 2.4. Sensor Placement and Hardware Selection

AE data acquisition utilized a PicoScope 4262 (Picotech, St Neots, UK), a 16-bit oscilloscope equipped with dual channels for digital AE measurement, as shown in Figure 4A. The AE events were recorded using an AE equipment, model AE B670 (QingCheng, AE Institute Co., Ltd., Guangzhou, China), as shown in Figure 4B. This resonant AE sensor features an integrated preamplifier. The design eliminated the need for an additional preamplifier, simplifying the wiring process. The GI150 sensor is particularly suitable for detecting flaws and cracks in metallic machinery and monitoring the structural integrity of pipelines, pressure vessels, bridges, and other structures of a similar nature. Furthermore, the GI150 can be used with the AE detectors of other manufactures. Before experimentation, the sensors were calibrated according to ASTM E1106 standards [47] to ensure signal reliability. AE sensors, provided by QingCheng, AE Institute Co., Ltd., Guangzhou, China, which are capable of detecting frequencies from 60 kHz to 400 kHz and operating in temperatures ranging from −20 °C to 50 °C, were employed. The PicoScope 4262 Data Acquisition Analog to Digital Converter (ADC), Picotech, St Neots, UK, with a bandwidth of 500 kHz, was configured with a sampling rate of 1 MS/s. This ADC converted high-frequency waves in the MZ crown into digital signal data. The data were then stored in a CSV file format and subsequently transferred to a computer for analysis. A coupling agent facilitated close sensor contact with the material via a waveguide to ensure effective signal capture. Calibration involved conducting conventional PLB tests on the MZ dental crown surface to introduce fracture energy, a method known for generating reproducible AE signals.

### 2.5. Acquisition of Acoustic Emission Data

The pencil lead fracture energy was introduced to the MZ crown. The tests utilized the same AE sensor (GI 150 AE, B670) by carefully breaking a 0.5 mm pencil lead against the surface of the dental crown at five different areas (the PLB test). The PLB test, also known as the Hsu–Nielsen pencil lead break [48], is a well-established technique to generate reproducible AE signals [49]. The PLB test was conducted on the MZ crown specimen using this consistent source (a Hsu–Nielsen pencil lead break similar to ASTM E976-84) [50] and the same data handling procedures. Measurements were taken at 500 points across five areas, totaling 2500 known positions (Hsu–Nielsen sources). In each position, the pencil lead was intentionally broken at least five times to capture adequate AE signals for calibration analysis, mainly focusing on the incisal area, as detailed in Table 1 and Figure 5. Across all classes, the peak amplitude values consistently ranged between 16 and 26 dB, with average amplitude values approximately centered around 20 dB. The methodology included the preprocessing and application of an Inception-ResNet-v2 CNN model.

### 2.6. Acoustic Emission Data Preprocessing

Subsequently, the oscilloscope, Picoscope 4262, Picotech, St Neots, UK provided the amplified signal data. These data were subjected to signal processing, which included denoising using a Bayesian filter. AE data preprocessing involved denoising with a Bayesian filter, with details of the denoising parameters provided in Table 2. Subsequently, the denoised signals from crack sites were transformed into 229 × 229 × 3 CWT images and fed into the Inception-ResNet-v2 architecture for further analysis. Each faulty class exhibited a distinct frequency pattern. A single dataset consisting of 262,143 data points was transformed into a scalogram image. Figure 6A displays plots of raw acoustic data collected from an AE-PBL setup for five faults. Figure 6B shows AE data involving Bayesian denoising; the other figure represents the CWT-scalogram image.

Our crack detection experiments were implemented using Python 3.12.5, USA on a Windows 10 64-bit operating system. The hardware included an Intel Core i9 processor (Santa Clara, CA, USA) with 10 cores and 20 threads, 128 GB of memory, an Nvidia GeForce Titan x1, and a Nvidia Tesla K80 and 1 GPU (Santa Clara, CA, USA). Scalogram-based images were obtained using Python in this work, and MATLAB R2023b, MathWorks, USA was employed for CNN training and classification. The dataset was randomly partitioned into 80% training and 20% test datasets for model development and accuracy evaluation [21,51,52,53]. Transfer learning was applied using a pre-trained Inception-ResNet-v2 model, with the best weights initialized. The training utilized the Adam optimizer with default parameters, modifying the top layers for a new softmax classification and output layer tailored to practical categories. In this study, each category contained 500 images with equal resolution, totaling 2500 images across five surface crack classes. The dataset was split into 80% for training (2000 images) and 20% for testing (500 images). Specifically, 400 images were used for training each surface crack category and 100 for testing. The training model (80% of datasets) was conducted over 2000 epochs using MATLAB R2023b, MathWorks, USA with an initial learning rate of 0.0001 and a minibatch size of 32. The CNN parameters were optimized by comparing training accuracy across iterations to achieve the most robust and accurate model.

### 2.7. Acoustic Emission Signal to Scalogram Processing

This tool is invaluable for analyzing signals featuring diverse scale phenomena, from gradual changes to sudden events. Scalograms in localizing high-frequency, short-duration events and low-frequency, long-duration events offer superior time and frequency resolution compared to spectrograms [54]. The CWT technique, integral to generating scalograms, enhances signal analysis by providing a detailed time-frequency domain, which is essential for fault diagnosis and DL-based fault classification models such as CNNs. A time-frequency representation of the signal, depicting energy density obtained through the CWT, is referred to as a scalogram. Scalograms are used to visualize signal components at different scales, from large to tiny fragments. The MATLAB wavelet toolbox, MathWorks, USA representation for various surface cracks was generated and is shown in Figure 7a–e.

### 2.8. A Prediction Model for the Crack Localization in the MZ Dental Crown

This study customized the Inception-ResNet-v2 model by modifying its fully connected layer. Initially designed for classifying 1000 classes with a fully connected layer of 1000 nodes, the proposed model added a layer comprising 5 nodes to categorize five specific classes (incisal, palatal, labial, right, and left). The architecture remained otherwise unchanged. The output layer employed a softmax activation function to assign probabilities to each class and determine the highest probability as the prediction. The deep learning framework, Inception-ResNet-v2, optimized for CNNs, processed input images sized 299 × 299 × 3 pixels from the CWT dataset. A global average pooling layer was utilized instead of a conventional fully connected layer, with a customized parameter setting, as specified in Table 3. The output layer of the network classified the fault statuses of MZ dental crowns into five distinct categories: incisal fault, palatal fault, labial fault, right fault, and left fault.

### 2.9. Performance Assessment Index

Four evaluation indices were used to fully evaluate the performance of the network in this experiment: accuracy, recall, precision, and F1 score. All states were identified from the crack test results: True Positive (TP), which represents the number of crack-true side samples correctly identified as crack-true; True Negative (TN), which represents the number of non-crack-true side samples correctly identified as non-crack-trued; False Positive (FP), which represents the number of non-crack-true side samples incorrectly identified as crack-true; False Negative (FN), which represents the number of crack-true side samples incorrectly identified as non-crack-true. The dataset was randomly divided into 80% for training and 20% for testing. Classification performance was evaluated using the following metrics: recall (sensitivity), which represents the number of correct optimistic predictions divided by the total number of positives; precision, which represents the proportion of true positives among all positive predictions; overall accuracy, which represents the number of correctly classified samples, divided by the total number of test samples; F1 score, which represents the harmonic mean of precision and recall, providing a balanced measure of accuracy [55]. The confusion matrix and F1 score were employed to evaluate the classification performance of the proposed scheme. The confusion matrix, a fundamental tool in classification, provided the basis for calculating metrics such as precision and recall. Recall was mathematically expressed by Equation (2), while precision was calculated using Equation (3). The F1 score, which measures classification accuracy by incorporating both precision and recall, was determined using the formula in Equation (4).
(2)$$Recall=\frac{TP}{FN+TP}$$
(3)$$Precision=\frac{TP}{TP+FP}$$
where *TP* represents true positives, *FP* represents false positives, and *FN* represents false negatives.
(4)$$F1-score=2*(\frac{Precision*Recall}{Precision+Recall})$$

### 2.10. On-Site Configuration for Experimental Studies

#### Flaw Generation in a Specimen

The crack system induced by Vickers harness indentations was examined by breaking specimens along a diagonal indentation [56]. To generate flaws, a MZ dental crown was kept dry and subjected to a digital microhardness test using a Shimadzu (MHT, HMV-2T, Kyoto, Japan). A diamond indenter with a load of 19.614 N and a holding time of 20 s was applied to the labial surface of the specimen and induced crack lines from the indent tips. Figure 8 displays the effect of the Vickers hardness test.

The AE sensor installation on the dental occlusion model, simulating field conditions for crack detection, is depicted in Figure 9a. The onsite experiment of the AE scheme with the pre-flawed dental crown under load is shown in Figure 9b. The data collecting module transformed the AE signals—caused by incisal contact and recorded by the AE sensor—into digital data. After pre-processing these digital data to eliminate noise, a DL algorithm, Inception-ResNet-v2, was used to identify flaws in the MZ dental crown.

## 3. Results

The performance of the deep CNN architectures for the five-class training was evaluated through supervised classification experiments on the proposed dataset. All models were trained on the training dataset, and performance was measured on the test set at 2000 epochs. Therefore, all performance metrics were referred to in the unseen test for a fair comparison. The performance metrics computed included classification total accuracy, recall, precision, F1-score, and AUC. The baseline experiment, shown in Figure 10 and Figure 11, involved the Inception-ResNet-v2 model with a target training dataset of crack regions’ images. In this experiment, 80% of the dataset was used for training and 20% for testing, with 2000 epochs, 313 min, and 27 s used for training.

Since precision, recall, and the F1-score depend on the positive class, their reported values are the results among the five classes (“true surface cracked” and “false surface cracked”). The results for the five training classes using all models are presented in Table 4, with the confusion matrix for the test set provided in Figure 10. The classification performance reaches 100% in all metrics using the Inception-ResNet-v2 models with data augmentation.

The confusion matrix in Figure 10 also shows the test performance of the Inception-Resnet-v2 network using AE-time series transformed into scalograms, with an overall test accuracy of 99.4%. Table 4 details precision calculations: incisal and left (100%), palatal, labial, and right (99.01%). The recall values were as follows: incisal (97%), palatal, labial, right, and left (100%). F1 score: incisal (98.477%), palatal, labial and right (99.502%), and left (100%). These high values—above 97%—indicate excellent classification results for the five AE-CWT data types used in the PLB-simulated cracking of MZ dental crowns.

The ROC AUC values were determined based on the complete set of predicted results shown in Figure 11. The ground truth test values and predicted probabilities on the test data were input into the function to calculate ROC AUC. The AUC provides an aggregate measure of performance across all possible classification thresholds.

### On-Site Dental Biting Using the Deep CNN Architecture

With a nondestructive single-sensor AE technique, coupled with a waveguide, Figure 12 shows the scalogram of the AE signals identified in the fault of the MZ dental crown under load conditions. The waveguide and AE sensor were used to locate cracks in the MZ crown. Prior to implementing the proposed AE sensor scheme under a load, the waveguide and AE sensor were first employed to localize cracks in the MZ crown, with these crack locations serving as references (i.e., actual results).

Following the AE parameter setting station, the waveguide is positioned on the MZ dental crown’s labial (anterior) surface to localize cracks (Figure 9). The suggested AE sensor technique is implemented under a load while preserving the identical dimensions and form that the setting station has previously examined. For every AE signal (from the MZ crown under load), the suggested DL approach locates the flaws in the dental crown by using the DL-generated probability value with the highest value. The crack localization results from the proposed DL-based AE sensor scheme are compared against the reference (actual results from flaw generation), described in Table 5.

In this research, the F1 score (classification accuracy) of the proposed AE sensor scheme for onsite localization of cracks in the labio-incisal area is 98%, compared to the fundamental flaw generation depicted in Table 6.

## 4. Discussion

This research specifically evaluated the Inception-ResNet-v2 architecture to analyze AE-CWT patterns, with the aim of enhancing crack detection and localization in MZ dental crowns. This study utilized CWT to generate 2D images from AE signals and found that the model performed well with minimal training data, provided that the data collection was well controlled. The results indicate that the model achieved low amplitude on simultaneous clinical model datasets, possibly due to inter-material bite contact. The incisal edge of the opposing tooth was restored with nanofilled composite (Filtek Z350, 3M ESPE), following findings by Kim R.J.Y. et al., who reported the highest AE events in Z350 [57]. The AE signals from biting were generated. Leveraging 2D contour maps and 3D coefficient projections, CWT of AE signals proved useful in identifying AE signal characteristics [58,59]. DL models, such as Inception-ResNet-V2, converged quickly and obtained excellent test results in fewer epochs, with network complexity being a significant factor in model accuracy [60]. The ability of the algorithm to detect and localize cracks with minimal input data and computation made it ideal for demonstrating the value of DL models in routine inspections, allowing for the automatic diagnosis of cracks in MZ dental crown structures. In the creation of flaws, limitations on physical clinical crowns with curvature necessitated the selection of a flat location. In this case, the labial area close to the edge of the tooth was selected. Accurate fractures in the labial and incisal of the crown portions were accurately detected by MicroCT (Skyscan 1272 Bruker—source: 100 kV to 100 µA, image pixel size = 11.999896 µm) using a camera pixel size of 8.96 µm. The 3D reconstruction of the MZ crown was performed with NRecon v2.2.0.6 software (Bruker, Billerica, MA, USA) after DL algorithm analysis, as shown in Figure 13.

The evaluation metrics, such as accuracy, precision, recall, and F1 score, were employed to assess the performance of the model. In the dental model testing, the Inception-ResNet-v2 model achieved 99.4% accuracy and an F1 score exceeding 98%, with minimal false positives and a perfect precision score of 1, highlighting its effectiveness in accurately predicting crack surfaces. The model demonstrated exceptional accuracy during the training and testing phases, with its derived features ensuring top performance. Our approach not only improved the precision of the model in crack detection and classification but also addressed the limitations identified in previous studies on dental material analysis. However, the algorithm could not analyze specific crack features such as breadth, length, and orientation. Recognizing the current data limitations, we would like to suggest that future research should focus on diversifying data collection techniques, increasing experimental data, comparing results to demonstrate the reproducibility and reliability of the PLB method in our specific application, comparing with other network model methods, implementing robust cross-validation methods, and exploring real-time clinical applications. These enhancements would be critical for increasing the versatility of the model across various clinical contexts and potentially surpassing the performance benchmarks established by previous research. The potential of the system extends beyond its current application, with prospects for development into a real-time defect detection system for various surfaces of zirconia dental crowns and eventual integration into IoT-enabled dental devices. The capability for early detection and autonomous identification of crack orientation is crucial for predicting failure causes such as those due to flexural or shear stress, underscoring the importance of early diagnosis and treatment in preventing crack propagation, both of which will ultimately lead to improved patient outcomes. Future research should also explore the incorporation of uncertainty estimation and incremental learning systems, which could further refine our approach and enhance the applicability of the model in diverse clinical settings.

## 5. Conclusions

This study introduced a novel approach for crack detection and localization in zirconia dental crown structures, utilizing the Inception-ResNet-v2 CNN model. Our method employed AE data, transformed through CWT, to create time-frequency images for deep learning analysis. This innovative application of CWT in dental materials marked a significant advancement, as it enabled the efficient integration of crack occurrence data into DL models. The model demonstrated high accuracy in crack detection and localization through careful hyperparameter optimization, which is a remarkable achievement, considering the complexities of data collection and the challenges of generating high-quality datasets in this field.

In conclusion, the findings of this study contribute to the existing knowledge in dental material classification and diagnosis and set the stage for future innovations in the field. By integrating AE-NDT testing with advanced DL techniques, we have developed a powerful tool for the early detection and identification of cracks in dental materials. The findings from this research have significant potential to influence clinical practice by offering enhanced diagnostic capabilities and ultimately improving patient care.

## Figures and Tables

**Figure 1 sensors-24-05682-f001:**
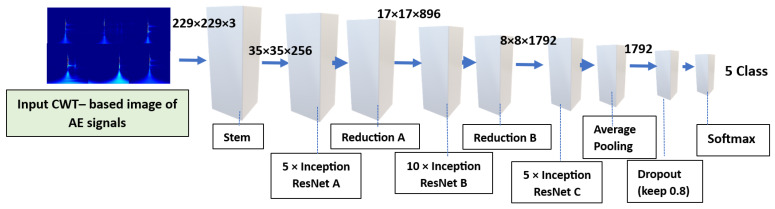
The architecture of the Inception-ResNet-v2 model.

**Figure 2 sensors-24-05682-f002:**
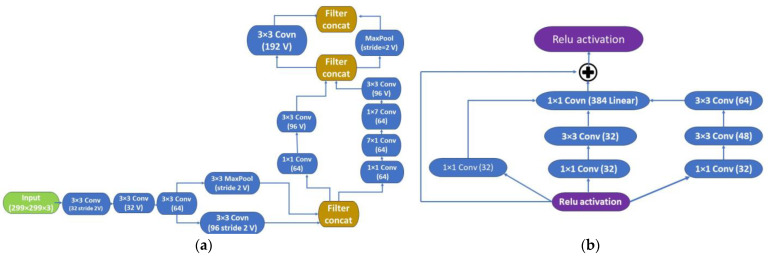
Inception-ResNet-v2 Network internal grid modules: grid sizes of 35 × 35, 17 × 17, and 8 × 8 in Stem (**a**); Inception-ResNet A (35 × 35 grid) (**b**); Inception-ResNet B (17 × 17 grid) (**c**); Inception-ResNet C (8 × 8 grid) (**d**); reduction modules A (35 × 35 to 17 × 17; k = 256, l = 256, m = 384, n = 384) (**e**); and reduction modules B (17 × 17 to 8 × 8 grid) (**f**).

**Figure 3 sensors-24-05682-f003:**
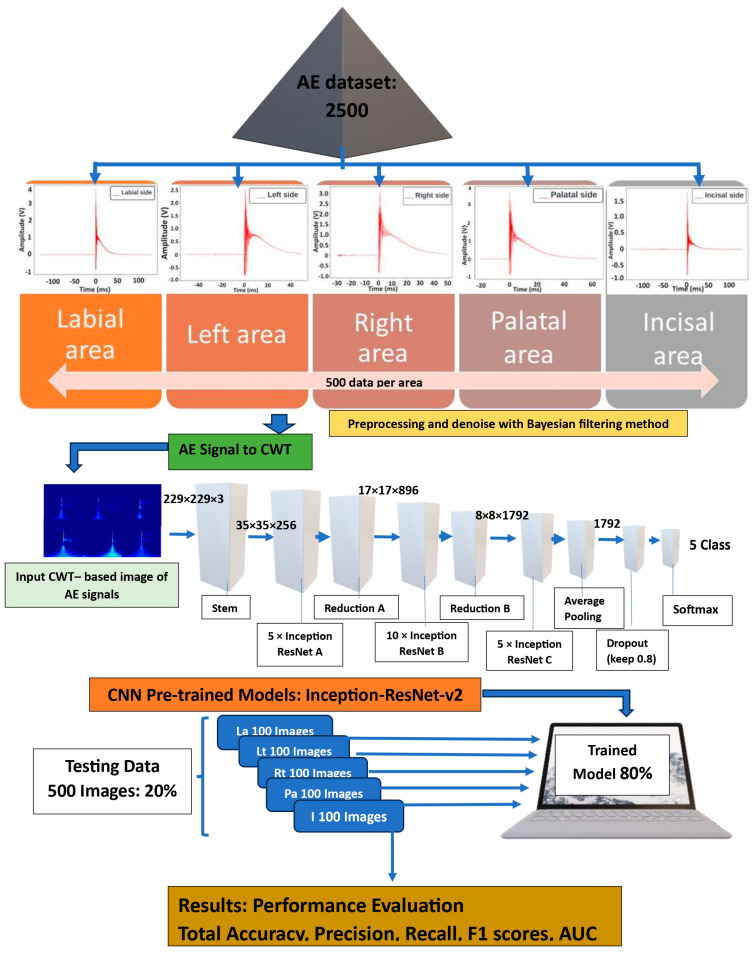
CNN application and methodology: Inception-ResNet-v2 architecture for classifying AE signals in MZ crown crack detection.

**Figure 4 sensors-24-05682-f004:**
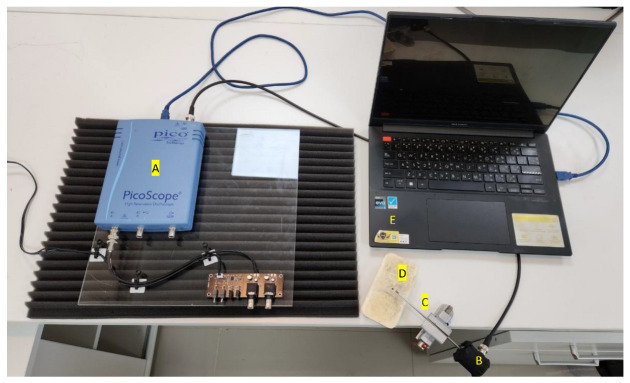
AE monitoring equipment: (A) picoscope 4262, (B) AE sensor (B670), (C) waveguide, (D) MZ dental crown, and (E) personal computer.

**Figure 5 sensors-24-05682-f005:**
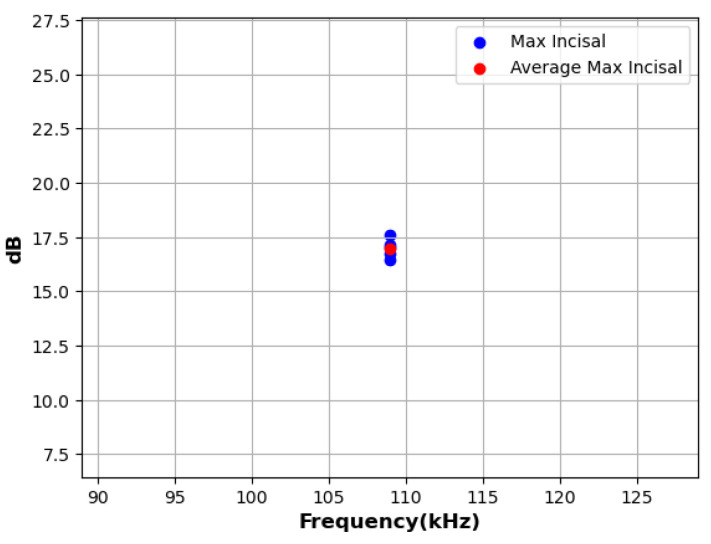
Depiction of the incisal class’s peak amplitude (dB) in relation to frequency (kHz).

**Figure 6 sensors-24-05682-f006:**
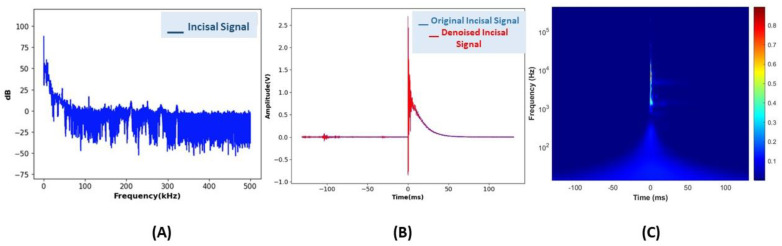
AE feature extraction: frequency-domain (**A**), time-domain signals (**B**), and scalogram transformation in the incisal area (**C**).

**Figure 7 sensors-24-05682-f007:**
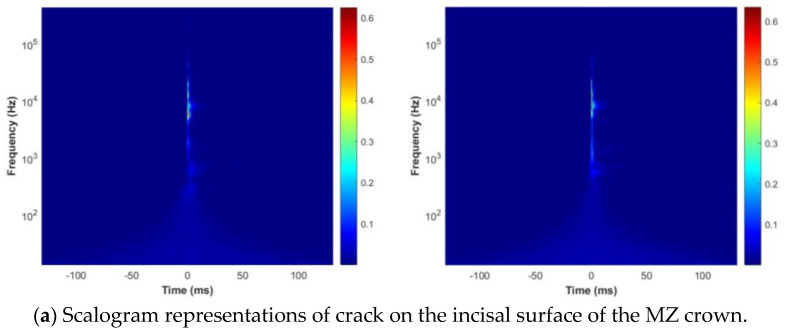
Scalogram depictions of cracks on incisal (**a**), palatal (**b**), labial (**c**), right (**d**), and left (**e**) surfaces of the MZ crown.

**Figure 8 sensors-24-05682-f008:**
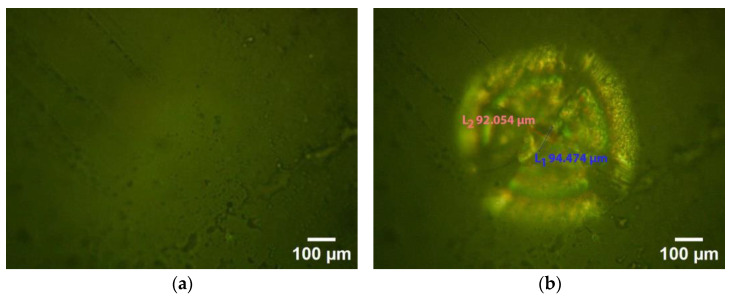
Surface before and after indentation: pre-indentation (**a**) and post-indentation crack formation (**b**).

**Figure 9 sensors-24-05682-f009:**
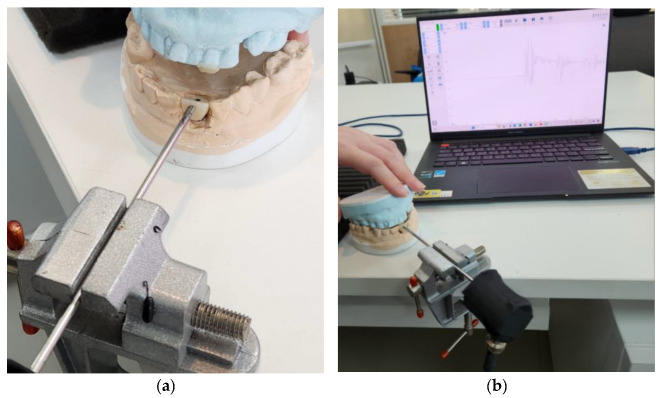
AE station for simulating dental biting: (**a**) AE sensor installation; (**b**) AE signal data from occlusal contact.

**Figure 10 sensors-24-05682-f010:**
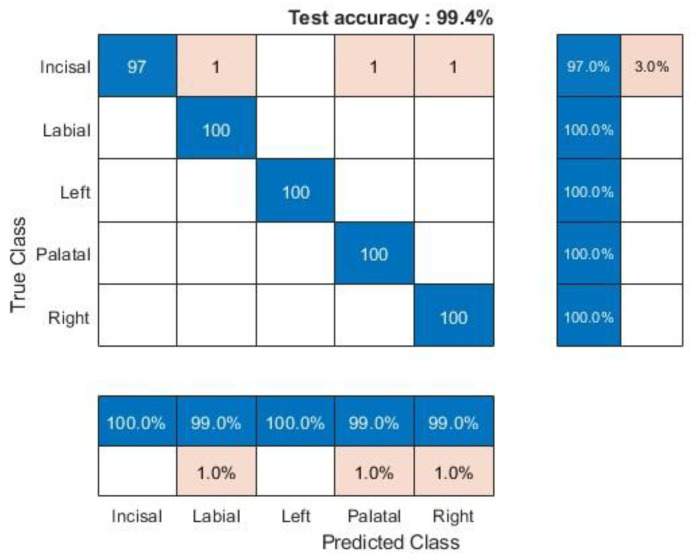
Confusion matrices for crack models on MZ crown surfaces using Inception-ResNet-v2.

**Figure 11 sensors-24-05682-f011:**
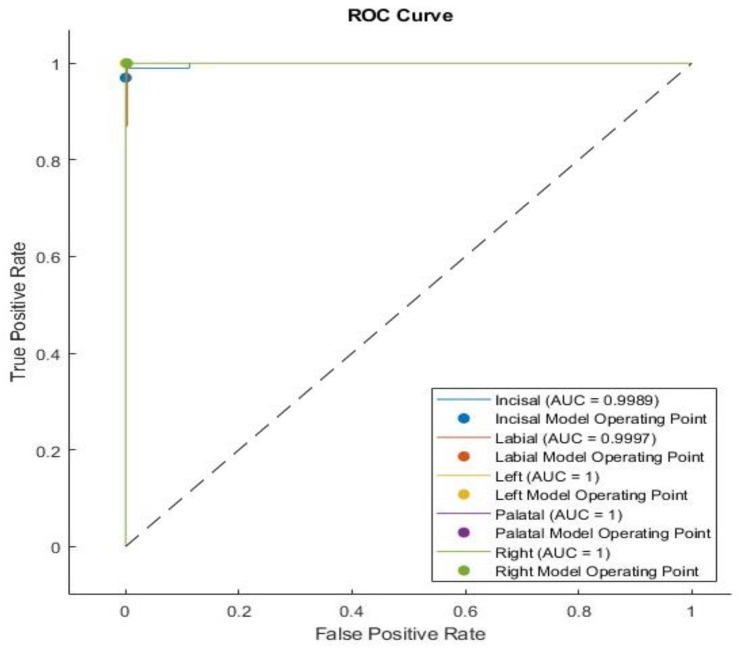
The ROC curves for the five crack tests show AUC values close to the ideal value of 1 across all cases.

**Figure 12 sensors-24-05682-f012:**
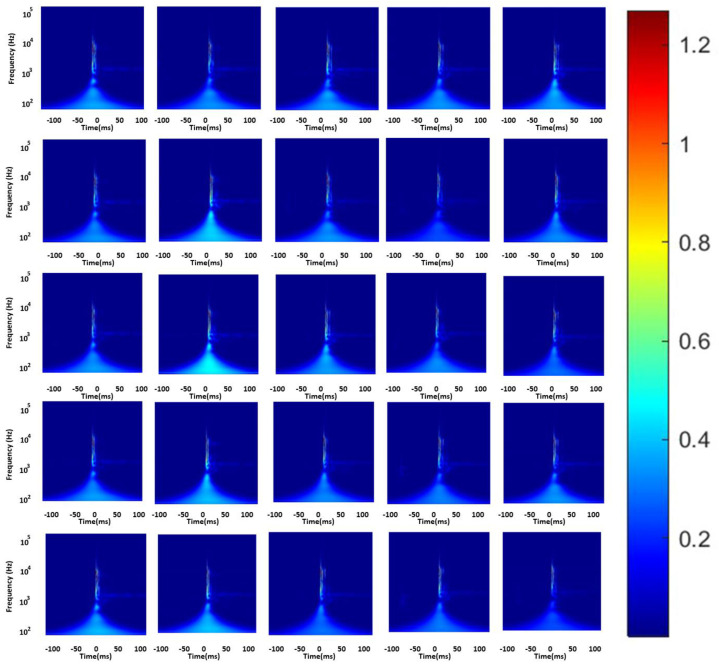
Evaluation of twenty-five scalograms using the Inception-ResNet-V2 architecture on an onsite dental model.

**Figure 13 sensors-24-05682-f013:**
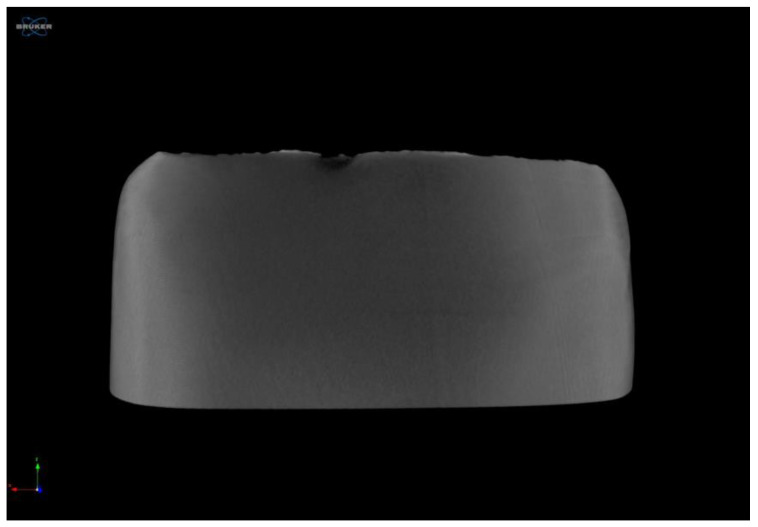
Three-dimensional micro-CT of MZ dental crown defect in the inciso-labial area.

**Table 1 sensors-24-05682-t001:** Peak and average amplitudes for the incisal class.

Sensor S/N	1st PLB	2nd PLB	3rd PLB	4th PLB	5th PLB	Average
B670	17.16	17.61	16.74	17.00	16.44	16.99

**Table 2 sensors-24-05682-t002:** Bayesian denoise parameter setting.

Parameter	Value
Level of Decomposition	8
The Symmetric Wavelet	sym4
Denoising Method	Bayesian
Threshold Rule	Median
Noise Estimate	Level Independent

**Table 3 sensors-24-05682-t003:** Parameters setting for the trained Inception-ResNet-v2 model.

Parameter	Value
Train/Test	80/20
Optimizer	Adam
Epoch	2000
Mini batch	32
Iteration per epoch	54
Initial Learning Rate	0.0001
Initial size	299 × 299 × 3

In the training and testing processes, L1-norm regularization was utilized to avoid overfitting due to excessive data points and divergence between the cross-entropy loss of training and testing datasets.

**Table 4 sensors-24-05682-t004:** The detection metrics obtained from evaluating image classification and crack surface detection architectures on the test set of the AE-CWT dataset.

Classes	Recall	Precision	F1 Score	AUC
Incisal	0.97	1	0.98477	0.9989
Palatal	1	0.9901	0.99502	1
Labial	1	0.9901	0.99502	0.9997
Right	1	0.9901	0.99502	1
Left	1	1	1	1

**Table 5 sensors-24-05682-t005:** Confusion matrix of Inception-ResNet-V2 for the real cracked simultaneous test.

Classes	Predicted True Crack	Predicted False Crack
Incisal	9	0
Labial	15	0
Right	0	1
Palatal	0	0
Left	0	0

**Table 6 sensors-24-05682-t006:** Analyzing image classification and crack surface detection on a test set of twenty data samples yielded the following detection metrics.

Classes	Recall	Precision	F1 Score
Inciso-labial	0.96	1	0.98

## Data Availability

The data presented in this study are available on request from the corresponding author.

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
