# Peer review of "Automated Crack Detection in Monolithic Zirconia Crowns Using Acoustic Emission and Deep Learning Techniques"

_sensors, 2024, doi:10.3390/s24175682_

Round 1
Reviewer 1 Report
Comments and Suggestions for Authors
This study aims to generate two-dimensional images of AE signals using CWT and analyse these images using a convolutional neural network (CNN) with the Inception ResNet-v2 architecture to achieve automatic detection and classification of cracks in monolithic zirconia crowns. This paper may be accepted if the following problems can be clarified:
1. Although the author mentioned in the introduction the early use of AE and DL models for dental material classification, further elaboration can be given on the results, shortcomings, and how this study improves and innovates on the basis of previous research.
2. There are some formatting errors in this manuscript, such as the misaligned section on the generation of defects in the sample on page 12. Please review the entire manuscript and address these issues throughout it.
3. The Inception-ResNet-v2 structure image in figure 1 and the flowchart in figure 3 are too low in resolution to show clearly. In figure 5, the name of the signal in figure A is ‘incisal signal’, while the signal in figure B is ‘incial signal’. It is recommended that the entire text be checked to further enhance the clarity of each image. In addition, the diagram in the full text should be aligned with the body.
4. The time-frequency horizontal and vertical coordinates are missing from the scale maps of five locations in figure 5(c) and figure 6, and ten scale maps corresponding to the other five locations are recommended for further elaboration.
5. In the Acoustic Emission Data Preprocessing, the data set is randomly divided into 8:2. Please explain why or add a theoretical reference.
6. In this paper, it is suggested to increase the comparison experiment, increase the comparison between the Inception-ResNet-v2 model in this study and other models, or compare it with the methods of VGGNet-19 and GoogLeNet Inception-v3 mentioned in the introduction.
7. It is suggested to analyse the feasibility of using pencil lead fracture (PLB) to simulate crack growth and to discuss whether the experiment can simulate the actual situation.
8. In this paper, the capability of the model is verified only by the test set, and it is hoped that cross validation can be further implemented to evaluate the performance of the model on different data subsets more comprehensively.
9. In the discussion section, the experimental results can be further analysed and discussed, such as the impact of different factors on the experimental results as well as the comparison with other relevant studies. For future research directions, more specific recommendations and perspectives could be put forward, such as how to further improve the accuracy and generalisability of models and how to apply this technique to actual clinical diagnosis.
10. Although the conclusion section has mentioned the high accuracy and potential applications of the model, it is recommended to add limitations to this study and provide prospects for future research directions. For example, factors such as dataset size, diversity, and control of experimental conditions may have an impact on the results. We can explore how to further improve the robustness and generalisation ability of the model to cope with cracks of different types, sizes, and directions.
Comments on the Quality of English Language1.Tense: scientific papers usually use the present tense to describe research results and conclusions, and the past tense to describe experimental processes and methods. Attention should be paid to full-text temporal consistency, especially when describing experimental methods and results.
2.Avoid redundancy: avoid unnecessary repetition of words or expressions. For example, “The model achieved an accuracy of 99.4667% , demonstrating that deep learning significantly improves The localization of cracks sites in MZ crowns” can be simplified to“The localization of cracks in MZ crowns” to avoid duplication.
3.Sentence structure: avoid overusing long or complex sentences, which may be difficult for the reader to understand. Try using simple sentences or complex sentences to express ideas clearly.
Author Response
Dear Reviewer1,
We would like to express our sincere gratitude for your valuable feedback and suggestions for improving our manuscript. We have carefully considered and addressed all your comments. A detailed point-by-point response to each of your suggestions is provided in the attached PDF file.
Thank you once again for your time and effort in reviewing our work.
Best regards,
Kuson Tuntiwong
Kuson.tu@kmitl.ac.th
Supan Tungjitkusolmun
Supan.tu@kmitl.ac.th
Pattarapong Phasukkit
Pattarapong.ph@kmitl.ac.th

Reviewer 2 Report
Comments and Suggestions for Authors
In this paper, the acoustic emission method and deep learning techniques are combined to realize the automated crack detection in monolithic zirconia crowns, which is an intresting topic. Before publication, some questations need to be explained.
1. Was this method first proposed by the authors used in crack detection in monolithic zirconia crowns? If not, similar or related research advances should be clearly given.
2. In practical application scenarios, teeth are tightly fitted together. How applicable is this method? How to eliminate the influence of other teeth?
3. Does the coupling agent have sufficient biosafety?
4. Where are the significant differences in scalogram representations of crack on the Incisal surface, palatal surface, Labial surface, Right surface of MZ crown?
5. If possible, validate the method in practical application environments.
Comments on the Quality of English Languageok
Author Response
Dear Reviewer 2,
Thank you very much for your valuable feedback and suggestions. We have carefully considered and addressed all of your comments. A point-by-point response to each of your remarks is provided in the attached PDF file.
We appreciate your time and effort in reviewing our manuscript.
Best regards,
Kuson Tuntiwong
Kuson.tu@kmitl.ac.th
Supan Tungjitkusolmun
Supan.tu@kmitl.ac.th
Pattarapong Phasukkit
Pattarapong.ph@kmitl.ac.th

Reviewer 3 Report
Comments and Suggestions for Authors
• In Introduction section, the authors mentioned that (At the moment, crack detection uses AE signals or transforms AE signals to continuous wavelet transform (CWT) for more accurate result interpretation.). Do you have any reasons or references for that statement considering the AE signal characteristics?
• In fig. 4, the frequency range of major signal features is over 100kHz. Considering the sampling frequency (200kHz), the reliability of the signal is suspicious.
• There are no axis scales for Figure5 C and CWT images.
• In Figure 7, there is no scale bar for the image.
• Why the authors adopted Inception-ResNet-v2 model? How about using N-layer CNN for the relatively which are more efficient for relatively small amount of data set?
• It is not clear whether the authors just used a monolithic zirconia plate for PLB test or acquired AE signals from crown specimens.
Comments on the Quality of English LanguageModerate
Author Response
Dear Reviewer 3,
Thank you very much for your valuable suggestions and recommendations to improve our manuscript. We have carefully considered and addressed all of your comments. A point-by-point response to each of your remarks is provided in the attached PDF file.
Once again, we sincerely appreciate your time and effort in reviewing our manuscript.
Kind regards,
Kuson Tuntiwong
Kuson.tu@kmitl.ac.th
Supan Tungjitkusolmun
Supan.tu@kmitl.ac.th
Pattarapong Phasukkit
Pattarapong.ph@kmitl.ac.th

Round 2
Reviewer 1 Report
Comments and Suggestions for Authors
1.There are a few formatting errors in the manuscript, such as the image on page 12 being out of alignment with the main text, and checking that the title text of each figure should not be too long.
2.This experiment uses pencil lead fracture (PLB) to simulate crack propagation. It is recommended to provide certain data or analysis to support the applicability of the PLB method in research.
3.In the experimental part, it is suggested to increase the comparison with other network model methods.
Comments on the Quality of English Language
It is recommended that the full-text language be checked for syntax errors again.
Author Response
Dear Professor,
I would like to express my sincere gratitude for the valuable time you have dedicated to reviewing and providing feedback on my work. Your insights and recommendations are invaluable in helping me develop and refine this piece to its fullest potential. Your detailed and astute comments have not only helped elevate the quality of the work but have also opened up new perspectives that I might have overlooked. Your guidance demonstrates your deep expertise and experience in this field. I am particularly thankful for your thorough and thoughtful suggestions, which will undoubtedly contribute to making this a more comprehensive and polished piece of work. Once again, thank you for your dedication and the care you've taken in providing such valuable feedback. I will carefully consider and incorporate your recommendations as I work to improve and finalize this project.
Sincerely,
Kuson Tuntiwong
Kuson.tu@kmitl.ac.th
Supan Tungjitkusolmun
Supan.tu@kmitl.ac.th
Pattarapong Phasukkit
Pattarapong.ph@kmitl.ac.th

Reviewer 2 Report
Comments and Suggestions for Authors
Accept.
Author Response
Dear Professor,
I would like to express my sincere gratitude for taking the time to review my manuscript and for providing such valuable suggestions. Your insightful feedback has been instrumental in helping me improve the quality of the work, and I am deeply appreciative of your efforts to ensure that the final product is as strong and polished as possible. Thank you once again for your dedication and support in guiding this work towards completion.
Sincerely,
Kuson Tuntiwong
Kuson.tu@kmitl.ac.th
Supan Tungjitkusolmun
Supan.tu@kmitl.ac.th
Pattarapong Phasukkit
Pattarapong.ph@kmitl.ac.th
Reviewer 3 Report
Comments and Suggestions for Authors
No suggestions
Comments on the Quality of English LanguageImproved
Author Response
Dear Professor,
I would like to express my heartfelt gratitude for your invaluable contribution in reviewing my work, as your insightful feedback and thoughtful recommendations have significantly enhanced the quality of my project. Your expertise and attention to detail have not only helped me refine my ideas but also opened up new perspectives that I had not previously considered. The time and effort you invested in providing such comprehensive and constructive criticism are truly appreciated, and I am committed to incorporating your suggestions to further improve and polish my work. Your guidance has been instrumental in elevating the standard of my research, and I am deeply thankful for your dedication to fostering academic excellence. Once again, thank you for your generosity in sharing your knowledge and for your role in helping me develop a more robust and well-rounded piece of work.
Sincerely,
Kuson Tuntiwong
Kuson.tu@kmitl.ac.th
Supan Tungjitkusolmun
Supan.tu@kmitl.ac.th
Pattarapong Phasukkit
Pattarapong.ph@kmitl.ac.th